# Distilling Estonian Text Domains for Production-Oriented Machine Translation

**Elizaveta Korotkova**     **Mark Fishel**
Institute of Computer Science
University of Tartu, Estonia
`{elizaveta.korotkova, mark.fisel}@ut.ee`

## Abstract

This paper explores knowledge distillation for multi-domain neural machine translation (NMT). We focus on the Estonian-English translation direction and experiment with distilling the knowledge of multiple domain-specific teacher models into a single student model that is tiny and efficient. Our experiments use a large parallel dataset of 18 million sentence pairs, consisting of 10 corpora, divided into 6 domain groups based on source similarity, and incorporate forward-translated monolingual data. Results show that tiny student models can cope with multiple domains even in case of large corpora, with different approaches benefiting frequent and low-resource domains.

## 1 Introduction

The quality of neural machine translation (NMT, Vaswani et al., 2017) systems heavily depends on training data and the text domains covered in it. Large-scale NMT Transformer models are usually trained on multiple corpora representing different domains (Kocmi et al., 2022), which in turn requires training models with higher representation capacity and an exceedingly large number of parameters, sometimes in the tens of billions for the largest models (Fan et al., 2020; NLLB Team et al., 2022).

However, using such models for inference in a production setting becomes more costly and cumbersome with increasing size. In parallel to the challenge of using more representational and learning power, a constraint from the practical side is to have models be as small and fast as possible for efficient deployment in production.

An additional challenge arises from the variability of natural language and different text domains and styles. While methods for training an NMT model to perform well on a particular type of text are relatively straightforward, the requirement of having a single NMT model translate multiple varieties of input text without a significant loss of quality on any of them due to interference between the domains in the training data remains more difficult.

In this paper, we aim to bridge the gap between previous research and systems applicable in production by experimenting with multi-domain knowledge distillation for NMT on the example of Estonian-English translation. We show that even for tiny NMT student-models and large-scale training data, it is efficient to train a single student model on data distilled by multiple fine-tuned domain-specific teacher models.

Our contributions are:

- we experiment with distilling the knowledge of multiple domain-specific teacher models within a single student model, focusing on very small student models;

- we use a sizeable parallel dataset of 18M sentence pairs, consisting of 10 corpora, which we divide into 6 groups based on similarity of their sources;

- we release our student models, test set translations, and generation code[1].

## 2 Related Work

**Knowledge Distillation for Machine Translation**  Knowledge distillation (Bucila et al., 2006; Hinton et al., 2015) is the technique of compressing the knowledge learned by a large model with high capacity and a large number of parameters or by an ensemble of models into a single smaller model. Knowledge distillation allows for

---

[1] `https://github.com/TartuNLP/multidomain-students`

increased speed and efficiency at inference time, while aiming to not sacrifice the quality of model performance to a significant extent.

Knowledge distillation methods were extended to the task of machine translation by Kim and Rush (2016). One of the methods they proposed is interpolated sequence-level knowledge distillation, consisting of several steps:

- a large teacher model is trained on a corpus of parallel texts;

- the teacher model is used to translate the source side of the parallel corpus into the target language;

- a smaller student model is trained using the original data as source and the teacher-generated (distilled) data as target.

In this way, the student model is trained with the goal of imitating the teacher model's probability distribution over the translations, thus constraining the task from the full space of natural language translations to the much smaller space of translations generated by the teacher, and making it more easily achievable for the small student model.

We follow the sequence-level knowledge distillation procedure proposed by Kim and Rush (2016) in our knowledge distillation experiments.

Recent advances in efficient MT have extended the limits of small and fast NMT models (Junczys-Dowmunt et al., 2018; Kim et al., 2019a; Heafield et al., 2021, 2022), using knowledge distillation, increasingly lightweight architectures and CPU optimization for faster inference, while suffering increasingly small quality decrease compared to full-scale models. However, these experiments are typically focused on single-domain or general-domain translation; we build on the findings of MT efficiency research and use them in a multi-domain scenario.

**Knowledge Distillation for Multi-Domain Machine Translation**   While training a neural machine translation model to perform reasonably well on one specific type of text is relatively straightforward, generalizing to multiple domains within a single model is more challenging. Typically, full-scale NMT models are trained on vast amounts of parallel data representing various text domains (Akhbardeh et al., 2021; Kocmi et al., 2022). Numerous methods which aim to improve

multi-domain MT performance have been proposed (Kobus et al., 2017; Tars and Fishel, 2018; Britz et al., 2017; Zeng et al., 2018).

The task of achieving good performance on multiple text domains, together with the need for fast and efficient translation, have lead to combining the methods of multi-domain neural machine translation and knowledge distillation.

Wang et al. (2019) focus on the task of multi-domain translation, using knowledge distillation for additional domain supervision: the probabilities (soft targets) produced by domain-specific models are used when training the unified multi-domain model.

Gordon and Duh (2020) adapt student models to one text domain at a time. They suggest distilling general-domain data to improve the performance of the general-domain student model, fine-tuning the best obtained model to in-domain data, and fine-tuning the teacher model to a specific domain and distilling this in-domain model. In our experiments, we follow Gordon and Duh in fine-tuning teacher models to domain-specific corpora and distilling them.

Our work shares the most similarities with Currey et al. (2020). Similarly to them, we fine-tune a general teacher to obtain several domain-specific teachers, which we then distill into a single student model. However, our work is closer to a real-world production scenario: we use a significantly larger training corpus which combines more individual parallel corpora from different sources, as well as much smaller student models, whereas Currey et al. train teacher models with 12 encoder and 12 decoder layers (roughly 100M model parameters), and their student models follow the Transformer-base configuration (6 encoder and 6 decoder layers, around 60M parameters).

Concurrently to Currey et al. (2020), Mghabbar and Ratnamogan (2020) also explore distilling several domain-specific teachers into a single student model, but use word-level instead of sentence-level knowledge distillation, and do not focus on decreasing the size of student models.

## 3   Methods

In our experiments, we aim to create neural machine translation models which 1) perform well on several data domains, and 2) are small and efficient.

To distill NMT models, we follow the sequence-

level knowledge distillation framework initially proposed by Kim and Rush (2016), where a smaller student model is trained using the synthetic target-side data produced by a larger teacher model, and experiment with distilling multiple domain-specific teacher models into a single student model.

We follow Currey et al. (2020) in employing a straightforward strategy:

1. train a general-domain teacher model,

2. fine-tune the teacher model to partitions of the data to obtain multiple domain-specific teacher models,

3. use the domain-specific teachers to forward-translate the data,

4. distill the domain-specific teachers into a single student model.

However, our student models are much smaller than the student models used by Currey et al. (2020), and we use significantly more data, bringing our setup closer to a full-scale real-world scenario.

### 3.1 Data

The experiments are performed on the Estonian-English language pair. We use 10 parallel corpora: Europarl (Koehn, 2005), JRC-Acquis (Steinberger et al., 2006), OpenSubtitles (Lison and Tiedemann, 2016), ParaCrawl (Esplà et al., 2019), EMEA, DGT, infopankki, GNOME, KDE4, and Ubuntu (Tiedemann, 2012). We divide the corpora into 6 groups as shown in Table 1. Europarl forms its own group of parliament proceedings texts (EU), EMEA a group of medical texts (MED), and OpenSubtitles a group of film and TV subtitles (SUBS). We merge the DGT and JRC-Acquis corpora into a group representing legal texts (LEGAL), ParaCrawl and infopankki represent texts crawled from the web (WEB), and, finally, GNOME, KDE4, and Ubuntu form the group of software localization texts (IT).

Table 1 also shows the number of sentence pairs in each group and corpus after cleaning; the total size of the parallel training corpus is ∼18M sentence pairs. The resulting corpus is highly unbalanced, with sizes of the groups varying from ∼100K examples for IT to ∼7.5M for WEB, which is realistic in a production scenario. From

each corpus, we separate a development set of 1000 sentence pairs and a test set of 500 sentence pairs. In addition to the held-out development sets, we also include the development split of WMT18 ET-EN set in the validation set. The test part of WMT18 ET-EN is used as an external test set.

### 3.2 Models

To train our models, we use the Marian framework (Junczys-Dowmunt et al., 2018). First, we train a teacher model from scratch using the 18M training data described above (this model is denoted as T-18M in Table 2). The teacher is a Transformer model, with shared SentencePiece (Kudo and Richardson, 2018) vocabulary of size 32,000 units, 6 encoder and 6 decoder layers, embedding dimension 512, feed-forward dimension 2048, 8 attention heads. The training is stopped when either BLEU (Papineni et al., 2002) or the mean word cross-entropy score on the validation set has not improved for 10 checkpoints, and the best checkpoint is chosen based on validation BLEU.

We then fine-tune the obtained teacher to each of the six data groups (the resulting domain-specific teachers are denoted, for example, T-EU or T-SUBS in Table 2). Fine-tuning is stopped when the validation metrics have not improved for 15 checkpoints. Next, we follow the interpolated sequence-level knowledge-distillation procedure (Kim and Rush, 2016): we forward-translate the parallel training data with the original general-domain teacher or with the corresponding fine-tuned teachers, generating 8-best lists for each source example. The best translation for each sentence is chosen based on its similarity to the original target sentence according to sentence-level BLEU.

In addition to the parallel data described above, we forward-translate 1M Estonian sentence pairs from the News Crawl corpus (articles from 2019 and 2020) and add those to the data the student models are trained on. (In this case, we cannot choose the translations which are closest to the original target, as no original target exists.) We also try fine-tuning the teacher model to these news data, where the target side was generated by the teacher itself.

Finally, we train several student models using the original source data and synthetic forward-translations obtained using the teacher models. For efficiency purposes, we follow Kim et al.

| group/corpus | corpus size | group size | domain |
|---|---|---|---|
| **EU** | | | |
|   Europarl | 593,637 | 593,637 | parliament proceedings |
| **LEGAL** | | | |
|   DGT | 2,241,448 | 2,637,222 | legislation |
|   JRC-Acquis | 395,774 | | |
| **MED** | | | |
|   EMEA | 211,722 | 211,722 | pharmaceutical documents |
| **SUBS** | | | |
|   OpenSubtitles | 6,868,517 | 6,868,517 | film & television subtitles |
| **WEB** | | | |
|   ParaCrawl | 7,601,013 | 7,614,325 | Web-crawled texts |
|   infopankki | 13,312 | | |
| **IT** | | | |
|   GNOME | 3,036 | 105,906 | software localizations |
|   KDE4 | 99,808 | | |
|   Ubuntu | 3,062 | | |
| **Total** | | 18,031,329 | |

Table 1: Sizes of corpora and corpus groups (number of sentence pairs) used for training the ET-EN teacher and student models, after cleaning

(2019b) and replace the self-attention mechanism in the Transformer encoders, which have 6 layers, with GRU-based cells, and use Simpler Simple Recurrent Units in the transformer decoders, which consist of 2 layers. The training is stopped if the metrics have not improved for 20 checkpoints. The resulting student models have disk size of 65 megabytes. S0 is the model trained using data produced by the initial teacher. S-FT uses the data forward-translated by the corresponding fine-tuned teacher for each of the corpora. S-FT-bal uses the same data, but after balancing the corpora: the total size of the training data is kept approximately the same as before, but each data group is downsampled or upsampled so that the sizes of all groups are approximately equal. The last model, S-ORIG, is trained for comparison on original (not forward-translated) parallel data.

We provide our S0, S-FT, and S-FT-bal student models, test set translations generated by them, and code used to generate and evaluate those translations at `https://github.com/TartuNLP/multidomain-students`.

## 4 Results

Table 2 shows the BLEU scores (Papineni et al., 2002) our teacher and student models achieve on the held-out and WMT18 test sets[2]. We can observe that fine-tuning on a data group noticeably improves the performance of the teacher model on held-out test sets within that data group. Not unexpectedly, the effect is more pronounced for smaller corpora, which are less represented in the whole corpus on which the original mixed-domain teacher is trained. We assume that the second important factor is the extent to which the corpus is narrowly specialized. For example, on the test set extracted from the very small and highly specific Ubuntu corpus, the performance of the fine-tuned teacher model is higher than that of the general teacher by a huge margin of 17.9 BLEU points, while the same performance gap is 3.1 points for OpenSubtitles and 1.9 points for Europarl.

It seems that fine-tuning the teacher on forward-translated monolingual data yields no positive effect. BLEU score on the WMT18 test set remains the same as for the general teacher, while scores on the held-out test set drop. This is not unexpected, as, while the teacher stops encountering data from other domains during fine-tuning, it also only encounters the data from the news domain that it forward-translated itself, and most likely cannot learn to exhibit any new behaviour on these data.

---

[2]sacreBLEU signature: `nrefs:1|case:mixed| eff:no|tok:13a|smooth:exp|version:2.3.1`

| group | EU | IT | | | LEGAL | | MED | SUBS | WEB | | NEWS | |
|---|---|---|---|---|---|---|---|---|---|---|---|---|
| corpus | Europarl | GNOME | KDE4 | Ubuntu | DGT | JRC-Acquis | EMEA | OpenSubtitles | ParaCrawl | infopankki | WMT18 | avg |
| T-18M | 40.9 | 33.8 | 29.7 | 37.8 | 44.2 | 57.5 | 46.8 | 31.9 | 50.5 | 30.9 | **30.4** | 39.491 |
| T-EU | **42.8** | 11.0 | 11.7 | 13.8 | 27.6 | 36.9 | 17.6 | 18.4 | 23.9 | 20.3 | 22.6 | |
| T-IT | 9.2 | **61.2** | **40.6** | **55.7** | 6.9 | 6.5 | 10.4 | 10.5 | 14.3 | 9.2 | 9.0 | |
| T-LEGAL | 29.4 | 17.1 | 12.7 | 18.3 | **49.7** | **64.6** | 24.3 | 8.1 | 23.8 | 16.9 | 16.5 | |
| T-MED | 9.6 | 9.4 | 7.3 | 8.4 | 12.7 | 15.4 | **66.3** | 4.4 | 10.7 | 7.3 | 7.2 | **48.727** |
| T-SUBS | 22.2 | 14.6 | 10.4 | 15.3 | 10.8 | 10.4 | 12.3 | **35.0** | 21.7 | 16.6 | 24.7 | |
| T-WEB | 38.2 | 29.5 | 23.5 | 33.0 | 36.4 | 50.2 | 39.4 | 24.9 | **52.3** | **37.4** | **30.8** | |
| T-NEWS | 37.7 | 24.3 | 20.9 | 27.1 | 28.8 | 38.4 | 29.5 | 28.7 | 35.4 | 27.0 | 30.4 | |
| S0 | **38.4** | 29.3 | **25.2** | 31.4 | 40.4 | 54.3 | 42.6 | **29.7** | 47 | 28.8 | 28.3 | 35.945 |
| S-FT | 38.3 | 29 | 25 | 31.6 | 41.3* | **54.5** | 41.3 | **29.7** | 48.6* | 30.6* | 28.5 | 36.218 |
| S-FT-bal | 38.2 | 45.6* | 23.7 | 45.3* | 38.8† | 50.9† | 49.3* | 27.3† | 41.6† | 25† | 26.2† | 37.445 |
| S-ORIG | 35.2 | 28.1 | 24.1 | 29.9 | 38.3 | 51.2 | 40.6 | 29.1 | 44.5 | 29.6 | 24.2 | 34.073 |

Table 2: BLEU scores of teacher and student models trained on 18M ET-EN sentence pairs as measured on different test sets. The columns represent the groups and corpora to which the test sets belong, and the rows indicate models. T-18M denotes the initial mixed-domain teacher model. T-EU, T-IT, etc. are teacher models fine-tuned on the corresponding groups of datasets. S0 is a distilled student model trained on texts forward-translated by T-18M. S-FT is a student model trained on data produced by the fine-tuned, domain-specific teacher models. S-FT-bal is trained on the same data as S-FT, but each data group is upsampled or downsampled so that all groups are of equal size, while the total number of training examples stays the same. S-ORIG is a model of the same configuration as the student models, but trained on original (not forward-translated) texts for comparison as a sanity check. The "avg" column shows each model's BLEU, averaged over all test sets (for fine-tuned teachers, we report a single average over the scores of each teacher's translations of test sets belonging to the corresponding group). Bold numbers indicate the highest BLEU scores for each test set among the teacher and among the student models. Individual test set results that show statistically significant improvements ($p \leq 0.05$) of S-FT and S-FT-bal in comparison to S0 are marked with *, while results that are significantly lower than S0 are marked with †.

While the behaviour of fine-tuned teachers is rather straightforward, the performance of student models is more varied. Comparing S0 and S-FT, we observe relatively similar performance: the difference on various test sets ranges from none (OpenSubtitles) to 1.8 (infopankki) BLEU points. The BLEU score averaged over all test sets is better for S-FT, but not by a very large margin. On the external WMT18 test set, S-FT performs best, although it only outperforms S0 by 0.2 BLEU points. On 6 test sets out of 11, S-FT is better than S0, although only on 3 of those the difference is statistically significant (Koehn, 2004), and on one more test set (OpenSubtitles) their result is the same.

The extremely small GNOME and Ubuntu corpora obviously benefit from balancing the data, and the scores on their test sets improve significantly compared to the unbalanced S-FT. Performance on the EMEA corpus, which comprises the second smallest data group, also noticeably benefits from upsampling. At the same time, if we compare the results obtained by S-FT and S-FT-bal on other corpora, we notice drops of 0.1-7.0 BLEU points.

The best average BLEU score is achieved by S-FT-bal, the student trained on data which is forward-translated by the fine-tuned teacher models and balanced.

## 5 Qualitative Analysis

Table 3 shows several example sentences from test sets belonging to each of the data groups, as well as their reference translations, translations generated by S0, S-FT, and S-FT-bal student models, and sentence-level chrF scores (Popović, 2015). In this section, we provide a brief description of varying model behavior on these examples.

In example 1, which comes from the Ubuntu corpus, only the model trained on balanced data manages to translate "ruutu soldat" as "jack of diamonds", while both S0 and S-FT translate "ruutu" literally ("squares"), and S0 translates "soldat" incorrectly ("solder" instead of the direct translation "soldier", which is likely due to subword interaction).

In example 2, the S-FT-bal model shows signs of overfitting: the content part of the sentence is identical to the reference, and the number "63" is generated at the start of the text, where the reference sentence has "53". However, there is no num-ber in the source sentence. Sentences produced by the S0 and S-FT models are, in fact, more exact translations of the Estonian source sentence ("you have the feeling"/"you feel" vs. "you think" and "the effect of Vivanza"/"Vivanza's effect" vs. "Vivanza").

In the infopankki example (3), all models manage to convey the original meaning of the source sentence, but S-FT-bal does so in a more informal style and with simpler grammar that the reference and the translations by S0 and S-FT (e.g. "work and business office" vs. "Employment and Economic Development Office", and "helps" vs. "Help is available"/"You can get help").

In the example from the OpenSubtitles corpus (4), all models use "his" instead of "her" (the Estonian pronoun "ta" does not have grammatical gender, so the correct English pronoun can only be inferred from wider context). The S-FT model uses the more informal contraction "it's", which is appropriate for the domain. The S-FT-bal model fails to translate a part of the compound word "kõnepost" and generates "voice" instead of the correct "voicemail".

In example 5 (Europarl), the S-FT-bal model is the only one not to use contractions ("We have" vs. "We've"), which do not typically occur in the formal style of parliament proceedings. However, all three model hypotheses are faithful.

The DGT example (6) sees the S-FT model translate very similarly to the reference, while both S0 and S-FT-bal overgenerate repetitively ("and ovens and ovens" and "Non-electric non-electric non-electric").

Finally, in the WMT18 example, all models fail to use the specific correct word "minesweeper", and instead translate the compound word "miini-jahtija" more literally as "mine hunter". Otherwise, the S-FT hypothesis is the only one to convey the full meaning of the source correctly.

## 6 Discussion

We observe that the fine-tuned teacher models predictably suffer from forgetting the general teacher's knowledge on domains other than the one the particular teacher is fine-tuned to. The extent of this forgetting varies, e.g. the teacher fine-tuned to Web-crawled text performs 2.7 BLEU points worse than the mixed-domain teacher on the Europarl test set, while the teacher fine-tuned to medical documents is 27.5 points worse on the

| | corpus | model | sentence | chrF |
|---|---|---|---|---|
| 1 | Ubuntu | SRC | ruutu soldat | |
| | | REF | jack of diamonds | |
| | | S0 | squares solder | 4.8 |
| | | S-FT | squares of the jack | 18.4 |
| | | S-FT-bal | the jack of diamonds | 95.0 |
| 2 | EMEA | SRC | Kui teil on tunne, et Vivanza toime on liiga tugev või liiga nõrk, pidage nõu oma arstiga. | |
| | | REF | 53 Tell the doctor if you think Vivanza is too strong or too weak. | |
| | | S0 | If you have the feeling that the effect of Vivanza is too strong or too weak, talk to your doctor. | 58.2 |
| | | S-FT | If you feel that Vivanza's effect is too strong or too weak, talk to your doctor. | 54.7 |
| | | S-FT-bal | 63 Tell the doctor if you think Vivanza is too strong or too weak. | 98.0 |
| 3 | infopankki | SRC | Töö otsimisel saab abi Töö- ja ettevõtlusbüroost. | |
| | | REF | The Employment and Economic Development Office provides help with your job hunting. | |
| | | S0 | Help is available in the Employment and Economic Development Office. | 60.6 |
| | | S-FT | You can get help in finding a job at the Employment and Economic Development Office. | 62.2 |
| | | S-FT-bal | The work and business office helps to seek the job. | 16.1 |
| 4 | OpenSubtitles | SRC | See on ta kõnepost. | |
| | | REF | It's her voicemail. | |
| | | S0 | This is his voice mail. | 52.6 |
| | | S-FT | It's his voice mail. | 67.1 |
| | | S-FT-bal | This is his voice. | 21.6 |
| 5 | Europarl | SRC | Oleme palju ära teinud, kuid töö ei ole veel läbi. | |
| | | REF | We have come a very long way, but the work is not yet complete. | |
| | | S0 | We've done a lot, but the work is not over. | 39.0 |
| | | S-FT | We've done a lot, but the job's not over. | 22.7 |
| | | S-FT-bal | We have done a lot, but the work is not over. | 44.4 |
| 6 | DGT | SRC | Mitte-elektriliste töötus- ja laboriahjude ja -põletuskambrite osad | |
| | | REF | Parts for non-electric industrial or laboratory furnaces and ovens | |
| | | S0 | Parts of non-electrical furnaces and ovens and ovens | 51.3 |
| | | S-FT | Parts of non-electric industrial or laboratory furnaces and ovens | 90.9 |
| | | S-FT-bal | Non-electric non-electric non-electric furnaces and oven parts | 46.8 |
| 7 | WMT18 | SRC | Sel poolaastal kuulub rahvusvahelise üksuse koosseisu ka Eesti mereväe miinijahtija Sakala. | |
| | | REF | This half-year, the Estonian minesweeper Sakala is also part of the international unit. | |
| | | S0 | This half-year is also part of the international unit Sakala, a mine hunter of the Estonian Navy. | 34.3 |
| | | S-FT | This half-year the international unit also includes the Estonian naval mine hunter Sakala. | 73.7 |
| | | S-FT-bal | In this half, the Estonian Navy mine hunter also includes the Estonian Navy mine hunter. | 63.7 |

Table 3: Examples of source-reference pairs from different test sets and corresponding translations produced by S0 (student model trained on texts forward-translated by a single mixed-domain teacher model), S-FT (student model trained on data translated by multiple fine-tuned teacher models), and S-FT-bal (trained on balanced data produced by multiple fine-tuned teachers) models. The last column shows sentence-level chrF score for each of the translations (sacreBLEU signature: `chrF2|nrefs: 1|case:mixed|eff:yes|nc:6|nw:0|space:no|version:2.3.1`).

OpenSubtitles test set than the original teacher, its score dropping to 4.4 BLEU, which suggests that these translations are not too far from random. The different levels of forgetting could potentially serve as a clue to domain similarity and guide the choice of manual data groupings.

There is a sizeable gap between the average performance of teachers fine-tuned to each data group and the best student model average. While the difference in capacity becomes even more drastic when we compare not one, but several large fine-tuned models to a single small student model, this gap remains large, and suggests the possibility of pushing the limits of student models' performance further.

Distilling the data clearly benefits the training of small models, the S-ORIG model lagging behind other student models. The best average BLEU is achieved by the student model trained on data distilled by multiple fine-tuned teachers and balanced between groups. However, in a real-world scenario trade-offs may still need to be made between the performance on specific domains, with average BLEU score not being representative enough for fine-grained evaluation.

## 7 Future Work

In our experiments, we used 10 corpora and, for fine-tuning, split them manually into 6 data groups based on the assumed similarity of their sources and topics. However, as demonstrated by Currey et al. (2020), the known domain labels may be suboptimal, and assigning the domains automatically can improve the multi-domain MT performance. Generating automatic domain labels using the general-domain model's internal data representations has been shown to further improve in-domain translation quality (Del et al., 2021). In future work, we would like to explore these methods for automatic domain discovery in conjunction with multi-domain knowledge distillation.

Aiming to bring our experiments closer to a production scenario, we tried incorporating forward-translated monolingual data into our multi-domain distillation setup. However, large-scale systems typically use back-translation and round-trip translation to increase the amount of training data. It currently remains unclear how to best incorporate monolingual data into the multi-domain knowledge distillation framework effectively, given the suboptimal results we achieved when fine-tuning

a mixed-domain teacher model to a forward-translated news corpus. We hypothesize that the teacher model cannot learn to exhibit any new behaviours when it is fine-tuned on data generated by itself. Thus, adding monolingual domains to distilled multi-domain systems is a potential topic for future exploration.

Another important direction for future work is extending our research to other language pairs and translation directions. While we perform our experiments on the Estonian→English language pair, which, to the best of our knowledge, has not been experimented with in a similar setting before, using other languages, especially low-resource ones, might lead to different results and insights.

## 8 Conclusion

In this work, we explored distilling multiple domain-specific neural machine translation teacher models into a single student model. While following procedures proposed in previous work, we incorporated research findings on model efficiency and focused on obtaining very lightweight student models. We used a training corpus of 18M Estonian-English sentence pairs, comprised of 10 unbalanced domains. We separated the domains into groups based on their perceived similarity, explored the effects of balancing, and incorporated monolingual forward-translated data into training of multi-domain students.

Our experiments show that the knowledge of several fine-tuned teachers models can be distilled into a very small student model, with balanced representation of domains further improving the average result. The massive total capacity of several fine-tuned teacher models has a huge average gain over the untuned teacher (almost 10 BLEU points) and the student models with their limited capacity achieve a much more modest increase in translation quality. Still, the increase in translation quality compared to the baseline student is stable and noticeable (+1.5 BLEU points).

## Acknowledgements

This work has been supported by the grant No. 825303 (Bergamot[3]) of European Union's Horizon 2020 research and innovation program. The authors thank the Unversity of Tartu's High-Performance Computing Center for providing

---

[3] https://browser.mt/

GPU computing resources (University of Tartu, 2018). We also thank the anonymous reviewers for their comments and suggestions.

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
