# OpenReview forum: "Distilling Estonian Text Domains for Production-Oriented Machine Translation"
_NoDaLiDa/2023/Conference — NoDaLiDa 2023_

### Official Review · Reviewer_xThi · 2023-02-19
**Easy to follow, reproducible work on multi-domain knowledge distillation for Estonian-English, but with miss-formulated contributions**

**Rating:** 7
**Confidence:** 5

**Review:**

The authors present their work on knowledge distillation in neural machine translation. They analyse the benefits of fine-tuning domain-specific (or corpora-specific) teacher models and forward-translating monolingual data to improve the quality of a smaller student model for Estonian-English. Their results show that such an approach allows for achieving a student model of better overall quality. The paper is easy to read and follow. The experiments are more or less well documented and could be reproduced by following the paper. However, I find what the authors state as their contributions confusing.

The authors state that they offer three contributions: 1) experiments by distilling multiple domain-specific teachers in one very small student model, 2) use of relatively large (for research) parallel data (18M sentence pairs), 3) incorporation of forward-translated data into knowledge distillation.

The first contribution is true only if we consider experiments on very small student models to be a substantial contribution. Prior work, e.g., Mughabbar et al. (2020), Currey et al. (2020) (which the authors cite) and Jooste et al. (2022) (which the authors missed citing) already published work on distilling student models using domain-specific teachers.

The second contribution is arguable since the authors are neither the first to work on 18M+ sentence pair datasets in NMT in general (e.g., see some industry papers describing their production systems or the Tatoeba Translation Challenge paper (Tiedemann, 2020)) nor the first to work on 18M+ sentence pair datasets analysing knowledge distillation (e.g., see Müller et al., 2020).

The third contribution is also arguable since the use of forward-translated monolingual data has also been investigated, e.g., see Xie et al, 2021 just as an example or Google Scholar to find more examples.

Having read what the authors believe their contributions are, I conclude that they are mostly untrue. I suggest the authors rethink what their contributions (to the field) are since the work does offer insights into the benefits of multi-domain knowledge distillation.

The forward-translated monolingual data in the authors' experiments lack an ablation study. I.e., what would the quality be without the forward-translated monolingual data?

117 - I believe that "during inference" is missing. Training knowledge-distilled models requires much more time... The only benefit is if the models are heavily used in production (i.e., during inference).

In summary, I believe the paper has merits and could be a good fit for the conference. The results can be insightful for researchers working on production systems and multi-domain systems. However, I believe the authors need to revise their contributions since most of what they consider contributions have been proposed/analysed before.

References:

Müller, M., Gonzales, A. R., & Sennrich, R. (2020, October). Domain Robustness in Neural Machine Translation. In Proceedings of the 14th Conference of the Association for Machine Translation in the Americas (Volume 1: Research Track) (pp. 151-164).

Xie, W., Hu, B., Yang, H., Yu, D., & Ju, Q. (2021, November). TenTrans large-scale multilingual machine translation system for WMT21. In Proceedings of the Sixth Conference on Machine Translation (pp. 439-445).

Mghabbar, I., & Ratnamogan, P. (2020). Building a Multi-Domain Neural Machine Translation Model Using Knowledge Distillation. In ECAI 2020 (pp. 2116-2123). IOS Press.

Currey, A., Mathur, P., & Dinu, G. (2020, November). Distilling multiple domains for neural machine translation. In Proceedings of the 2020 Conference on Empirical Methods in Natural Language Processing (EMNLP) (pp. 4500-4511).

Jooste, W., Haque, R., & Way, A. (2022). Knowledge Distillation: A Method for Making Neural Machine Translation More Efficient. Information, 13(2), 88.

Tiedemann, J. (2020, November). The Tatoeba Translation Challenge–Realistic Data Sets for Low Resource and Multilingual MT. In Proceedings of the Fifth Conference on Machine Translation (pp. 1174-1182).



**Paper Type:**

Long paper

---

### Official Review · Reviewer_8Eke · 2023-03-01
**Distilling Estonian Text Domains for Production-Oriented Machine Translation**

**Rating:** 7
**Confidence:** 5

**Review:**

The paper presents experiments with the distillation of domain-specific teacher models to a multi-domain and compact student model in the domain of neural machine translation. 10 different corpora were divided into 6 groups to fine-tune teacher models into domain-specific translation models and the multi-domain student is created using sequence-level distillation on synthetic training data produced by the teacher models. The experiments are conducted on Estonian-English translation tasks.

The setup si straightforward and uses standard pipelines for distillation of translation models. The innovation is the use of multiple domain experts as teachers to see whether their knowledge can effectively be injected in a tiny student model that covers all domains. The paper is well written and the experiments are convincingly presented and discussed. The results are promising and the benefits are very clear.

The teacher models are rather modest in size and i wonder whether it would actually be beneficial to have stronger teachers with larger architectures with this amount of training data. It is also quite common to use ensembles of teachers to improve the teacher material for student training even stronger. This should be explored in future work. Some initial discussion about this could probably be added to this paper already.

The student uses a GRU RNN as the encoder, which I find a bit surprising. Did you also try a transformer-based encoder instead to see what the impact of the encoder architecture is on final performance. Reducing the decoder to more shallow RNNs is quite common but for the encoder I am not sure whether there could be quite some impact especially for longer sequences.

In one of the settings, the authors also tried to fine-tune a model with forward-translated data. Unsurprisingly, this fails, and I actually wonder about the motivation of this approach. It really feels like some kind of self-learning that is doomed to fail and I don't really see the motivation. This must lead to some heavy overfitting and probably also catastrophic forgetting and I wonder whether there is some potential benefits that I miss. In the discussion section, there is also a bit of strange discussion on this setup in connection with the difficulty of using monolingual data in distillation. I don't see the connection and would say that those questions are very different things. Mixing forward-translations with back-translations in distillation would not relate to fine-tuning teachers on forward translated material in my opinion.

In the setup, I actually wonder whether you also considered to add domain-labels in the multi-domain student to make it possible to learn to generate different domain data. Do you think this would work and make the student model more flexible in coverging different domains?

The paper is well done, maybe a bit preliminary and I wonder whether it could be turned into a short paper. It is now a bit more than 6 pages but some things can be condensed and repetition avoided. Also the table takes a full page which could be optimized.

**Paper Type:**

Long paper

---

### Official Review · Reviewer_AvkR · 2023-03-10
**Distilling Estonian Text Domains for Production-Oriented Machine Translation**

**Rating:** 7
**Confidence:** 4

**Review:**

General remarks: In general this is a well written paper. It describes the experiment well and shows that it can be useful. Regarding testing on all these datasets, I'm not so sure. In some cases these evaluation sets they must be very small. Is all of this meaningful? Wouldn't it get the point better through just to use one or two evaluation sets and try to understand what is happening and describe that?


Other remarks:

36: What is a full-scale NMT transformer model? Define!

41: Sometimes in the tens of billions: This is just correct for very large language models suchs as GPT-3 and larger. GPT-2 was 1.5 billion. T5 11 billion. Multilingual BERT is 600 million ...

291: Cite SentencePiece paper

295: Cite BLEU paper when you first talk about BLEU.

360: Why 20 checkpoints and not 10 or 15 as before? How is the number of checkpoints chosen in each case?

415-418: The Ubuntu corpus is tiny. How is this huge margin possible? May it be because the evaluation data is very small and contains sentence pairs that are nearly identical to training data? What is the statistical significance? I would guess this isn't meaningful at all, when you take a better look at it. But further information is needed and a better discussion.

528-530: Highest BLEU scores in bold. But are they statistically significant? You should calculate statistical significance for this (paired bootstrap resampling - easy with SacreBLEU, and report SacreBLEU signatures for this to be easily comparable).

577-584: Significance testing needed.

References: Check these arXiv papers again, some of them have probably been published at different venues since.

**Paper Type:**

Long paper

---

### Decision · Program_Chairs · 2023-03-17

Accept